# Physical Activity, Sedentary Behaviour and Cardiovascular Risk Factors in Overweight Low-Income Schoolchildren: A Complex System Perspective

Thaynã Bezerra [1,2,*], Anastácio Souza Filho [1], Natália Quirino [3], Paulo Bandeira [1], Luciana Cabral [4], Cézane Reuter [5], Clarice Martins [4,6] and Ferdinando Carvalho [7]

1 Physical Education Department, Regional University of Cariri, Pimenta, Crato 63100-500, Ceará, Brazil; anastaciosouzafilho@gmail.com (A.S.F.); paulo.bandeira@urca.br (P.B.)
2 Medicine Department, University Center Paraiso, Araripina 56280-000, Pernambuco, Brazil
3 Medicine Department, Nova Esperança Faculty, João Pessoa 58010-000, Paraíba, Brazil; natalia_mesquita@hotmail.com
4 Physical Education Department, Federal University of Paraiba, João Pessoa 58010-000, Paraíba, Brazil; luciana-gatto@hotmail.com (L.C.); claricemartinsufpb@gmail.com (C.M.)
5 Physical Education Department, University of Santa Cruz do Sul, Santa Cruz do Sul 96810-012, Rio Grande do Sul, Brazil; cezanereuter@unisc.br
6 Research Centre of Physical Activity, Health and Leisure, Faculty of Sport Sciences, University of Porto, 96810-012 Porto, Portugal
7 Physical Education Department, Federal University of Vale do São Francisco, João Pessoa 48903-435, Paraíba, Brazil; ferdinando.carvalho@univasf.edu.br
* Correspondence: thaynaalves.ef@gmail.com; Tel.: +55-88-3102-1204

**Abstract:** Background: This study analysed through the perspective of networks the association between physical activity (PA), sedentary behaviour (SB), and cardiovascular risk factors in overweight low-income schoolchildren from the perspective of complex systems. Methods: The sample consisted of 41 overweight children between 6 and 10 years old (56% girls). PA and SB were analysed by accelerometer, and the measures of cardiovascular factor risk were: anthropometric and hemodynamic measurements, body fat, lipid profile and glucose, cardiorespiratory fitness (CRF), left ventricular mass (LVM), and Sokolow–Lyon. Network analysis with Bootstrap-1000 was performed to analyse the association between PA, SB and the cardiovascular risk factors. Results: Moderate-to-vigorous PA was positive related to waist circumference (WC; 0.499), HDL-C (0.307), and CRF (0.276), and negatively associated with BMI (−0.251) and Fat (−0.341). For SB, positive associations were seen with WC (0.326), CRF (0.296), LVM (0.250) and Sokolow (0.215). In addition, the centrality indicators highlighted WC as the most important variable in the network. Conclusion: Interventions that aim to mitigate the harmful effects of obesity on cardiovascular risk factors in overweight children should consider WC as an important variable in the system.

**Keywords:** cardiovascular factors; network analysis; obese children; physical activity; sedentary behaviour

## 1. Introduction

Overweight and obesity in childhood have a complex etiology, involving the simultaneous non-linear interactions between genetic, biological, environmental and behavioural factors [1]. Children from low-income settings show a high prevalence of obesity [2] and are more likely to develop cardiovascular risk factors, such as decreased glucose tolerance and type 2 diabetes mellitus, high lipid profile, high systolic blood pressure and atherosclerosis [3,4]. Furthermore, overweight children tend to have greater left ventricle mass (LVM) hypertrophy, when compared to their normal-weight peers [5]. LVM is a dysfunction that appears even before other cardiovascular risk factors, such as arterial hypertension. Despite

being considered the best indicator of a coronary lesion, the analysis of LVM has still been neglected in children [6].

It is also known that children's movement behaviours, such as low levels of physical activity (PA), and high time spent in sedentary behaviour (SB), are associated with cardiovascular risk factors' incidence, and tend to be more prevalent in overweight and obese children [4] living in socially vulnerable contexts. Studies point out that in middle-income countries such as Brazil, low socio-economic status is related to lifestyle determinants for obesity once it hinders access to healthy foods and PA opportunities [7,8]. Moderate-to-vigorous physical activity (MVPA) is related to improvements in cardiorespiratory fitness (CRF) (Ross et al., 2016) and acts as a protective factor against cardiovascular diseases [9]. Conversely, results of a recent review study show that SB, assessed through screen time, negatively influences outcomes related to cardiovascular health and childhood obesity [10], though further studies using objective measures of SB are needed.

In fact, movement behaviours such as MVPA and SB are interrelated [11]. Moreover, the different cardiovascular risk factors are also interrelated [12,13] and can cluster in the same person [14]. Therefore, factors involved in obesity´s etiology, such as movement behaviours and cardiovascular risk factors, should be seen as a dynamic, non-linear and integrated system. Considering that the presence or absence of a single variable within a model composed of interrelated variables may completely change its nature [15], from a theoretical and statistical perspective, the possible dynamic relationships between movement behaviours and cardiovascular risk factors might be explored through a network perspective, taking into account its non-linearity [16]. Thus, the present study aimed to analyse the relationship between MVPA, SB, and cardiovascular risk factors in overweight low-income children through a network approach.

## 2. Materials and Methods

### 2.1. Study Description

This cross-sectional study used baseline data from the school-based program "Panpes", which aimed to analyze the effects of a multicomponent intervention on health and behavioural outcomes of obese children. All the Helsinki Declarations' ethical aspects were followed (Association, 2013). The evaluation methods and procedures were approved by the Research Ethics Committee of Health Science Center (protocol n. 1.311.598) and by the Education Board of the city.

### 2.2. Sample

Eligibility criteria for schools' and children´s selection were established as follows:

- School:
  - Elementary public schools from central and suburban and deprived areas from Petrolina-PE, Brazil;
  - Indoor gym and with a minimum of one hundred students;
  - Overweight children (85th percentile for sex and age, according to the World Health Organization [17]).
- Children:
  - Overweight or obese [17];
  - Properly enrolled in participating schools;
  - Parental consent to participate.

Afterwards, the number of children who should have their weight and height measured, in order to identify excessive body weight (defined by the 85th percentile for sex and age, according to the World Health Organization (WHO) [17]), was calculated by using the number of children who met the inclusion criteria (2.231) multiplied by 45.0%, which was the prevalence of overweight and obesity in the Northeastern Region of Brazil. This resulted in a sample of 970 children, who were equally distributed (random selection of 122 children

per school) in the eligible schools and grades. From the total of 970, 194 children presented excessive weight, corresponding to 20% of the sample.

To determine the minimum sample size, a power of 0.80, effect size 0.3, error of 5% and an increase of 25% for possible losses and refusals were considered, totaling 30 children by group. Two schools met the inclusion criteria. All parents of overweight children found in both schools were invited to participate in the study (60 children in all).

### 2.3. Data Collection

Data were collected in February 2016 by five physical education teachers and three cardiologists, who were previously trained and supervised by the project´s coordinator. Data were collected at the school (anthropometric, hemodynamic measurements, lipid and glucose profile, cardiorespiratory fitness, physical activity and SB) and the clinic of cardiology (left ventricular mass) during the weekend, in moments previously scheduled with the parents and the school board.

- Anthropometric measurements

Body mass (kg), body height (cm) and waist circumference (WC; cm) measurements were assessed following WHO standardisation [17]. Body mass was measured using a Welmy® (São Paulo, Brazil) digital scale with 100 g accuracy. To measure body height, a stadiometer attached to the scale was used. The vertex and the plantar region of the children were used as reference points, and body mass index (BMI) was calculated. Children who presented BMI above the 85th percentile for age and sex were considered overweight [17]. In all measurements, children were standing in bare feet and wearing light clothes.

WC was assessed at the midline between the lower edge of the rib cage and the upper limit of the iliac crest and was measured with Sanny® (São Paulo, Brazil) metal tape measuring 1 mm [17].

The sum of skinfolds (triceps and subscapular skinfolds) was performed according to the standardisation protocol proposed by Harrison et al. [18], using a Lange® (Seko, USA) skinfold caliper.

All anthropometric measurements were performed in triplicate, by the same evaluator. For analysis, the mean value was adopted.

- Hemodynamic measurements

Systemic blood pressure was measured by a single evaluator, using a digital automatic device (Omron®; Osaka, Japan); this equipment has good levels of sensitivity and specificity for blood pressure measurements in children and adolescents [19]. Three measurements were performed, with a five-minute interval. Measurements were always obtained under similar conditions, in the same place and position, with the left arm being raised to the height of the midpoint of the sternum and resting on a table, according to the Brazilian Society of Cardiology [20], and the median value was used for analysis. This procedure has already been adopted in previous studies [21]. Mean blood pressure (MBP) was considered for the analysis; the same was calculated by ((2 x diastolic blood pressure) + systolic blood pressure)/3).

- Lipid and glucose profile

For biochemical profile measurements, 10 mL of venous blood was collected from the antecubital vein after 12 h of fasting. The blood sample was deposited in a vacuum tube with separating gel and without anticoagulant and was stored in thermal boxes for later analysis in a specialised laboratory. Triglycerides (TG), total cholesterol (TC), low-density cholesterol (LDL-C), high-density cholesterol (HDL-C) and glucose levels were determined in mg/dL [22].

- Cardiorespiratory Fitness

Cardiorespiratory fitness (CRF) was verified through the 6 min running and walking test proposed by Gaya et al. [23]. The test was conducted on the school court and was performed individually. The test was guided by a Physical Education teacher, who monitored

the time and verified the distance reached by the child during the given time. During the test, motivational words were directed to the children.

- Left Ventricular Mass

Left ventricular end-diastolic and end-systolic measurements were obtained with the patient in a partial left lateral position according to recommendations by the American Society of Echocardiography [24]. Frames with optimal visualisation of interfaces and showing simultaneous visualisation of the septum, left ventricular internal diameter, and posterior wall were used. A Level 3 echocardiographer performed the interpretations. Left ventricular mass was calculated by using the Devereux formula: left ventricular mass (g) = 0.80 × 0.6 g. The left ventricular mass was indexed according to body surface area.

The voltage based on the Sokolow–Lyon criteria has been used to diagnose left ventricular hypertrophy and has been associated with the presence of cardiovascular disease [25]. Individual leads were analysed by measuring the tallest R or R′ and the deepest S or QS complex in all the precordial and using the PR segment as baseline. In cases of voltage differences within the same lead, only the largest complex was selected. The proposed criteria were obtained by adding SD to the in V4 (SD + SV4). The Sokolow–Lyon voltage was obtained by adding the amplitude of S in V1 and the amplitude of R in V5 or V6 ≥ 3.5 mV (SV1 + RV5 or RV6); the limb lead voltage criteria amplitude of R in aVL > 1.1 mV (RaVL) and amplitude of R in L1 > 1.4 mV (RL1).

- Physical activity and Sedentary Behaviour

PA and SB were objectively measured using accelerometry (GT3X+, ActiGraph®, Pensacola, USA). Children were instructed to use the accelerometer affixed to the waist during eight consecutive days and remove for fighting and water activities, and at night [26].

Data were collected at a sampling rate of 80 Hz and epochs of 15 s. To reduce the data, the ActiLife 6.12 software was used, and data validity was set adopting the following criteria: three weekdays, and one weekend day; at least 10 h/day of registered data (06:00 to 00:00); 10 or more consecutive minutes of zero counts were established as non-wear time [27]. The time spent in SB and MVPA were obtained by the weighted average of minutes/day, using established cut-off points proposed by Evenson et al. [28].

- Data management and statistical analysis

To characterise the sample, a Student's *t*-test was used to compare the means of all variables between boys and girls.

A network analysis was conducted to assess the associations between MVPA, SB, and cardiovascular risk factors. The Fruchterman–Reingold algorithm was applied, and data were shown in a relative space in which variables with strong permanent statistics are together, and with weak applied variations repelled one another [29]. The least absolute contraction and selection operator was used to obtain regularisation and to obtain a less-sparse model [30]. The partial correlation parameter was adjusted to 0.25 to create a network with greater parsimony and specificity [31]. Additionally, a 1.000-bootstrap resampling was performed.

Three centrality indicators were used to determine the role of each variable in the emerged network: (1) betweenness (centrality between the parts), estimated from the number of times that a node is part of the shortest path among all the other pairs of nodes connected to the network; (2) closeness, which is determined from the inverse of the distances from one node to all others; and (3) strength (degree/centrality), which is the sum of all the weights of the paths that connect a node to the others [32]. The Jasp software version 0.12 was used.

## 3. Results

Sixty children were invited to participate in the study, of which nineteen were not included (did not present parental consent or did not attend assessments). Thus, 41 children (56% girls; 7.85 ± 0.96 years old) were evaluated. Boys were more active and less sedentary

than girls, as they spent more than 10 min/day on MVPA and less than 60 min/day on SB (Table 1).

**Table 1.** Sample characterisation.

| Variables | Boys (n = 18) (Mean ± SD) | Girls (n = 23) (Mean ± SD) | p | d' Cohen |
|---|---|---|---|---|
| Age | 7.7 ± 0.9 | 8.0 ± 1.0 | 0.277 | −0.347 |
| MVPA (min/day) | 34.6 ± 16.4 | 24.6 ± 13.5 | 0.037 * | 0.679 |
| SB (min/day) | 330.4 ± 44.6 | 389.0 ± 68.6 | 0.003 * | −0.987 |
| BMI (kg/m²) | 21.7 ± 3.2 | 22.1 ± 2.6 | 0.621 | −0.157 |
| WC (cm) | 68.8 ± 3.2 | 22.1 ± 2.6 | 0.822 | 0.071 |
| Fat (%) | 34.2 ± 9.2 | 34.7 ± 6.5 | 0.841 | −0.063 |
| MBP (mmHg) | 84.5 ± 6.9 | 80.3 ± 8.2 | 0.091 | 0.546 |
| TC (mg/dL) | 159.2 ± 35.3 | 156.6 ± 35.4 | 0.800 | 0.080 |
| HDL-C (mg/dL) | 37.4 ± 3.8 | 37.3 ± 7.8 | 0.962 | 0.015 |
| LDL-C (mg/dL) | 101.9 ± 29.8 | 97.8 ± 25.7 | 0.635 | 0.151 |
| TG (mg/dL) | 96.3 ± 45.1 | 100.9 ± 52.3 | 0.769 | −0.093 |
| Glucose (mg/dL) | 82.4 ± 5.8 | 82.7 ± 6.2 | 0.891 | −0.044 |
| CRF (meters) | 748.6 ± 73.7 | 752.0 ± 95.3 | 0.901 | −0.039 |
| LVM (g) | 49.9 ± 9.4 | 48.2 ± 12.4 | 0.629 | 0.153 |
| Sokolow–Lyon | 33.4 ± 6.9 | 31.7 ± 5.6 | 0.384 | 0.277 |

MVPA = Moderate to vigorous physical activity; SB = Sedentary behaviour; BMI = Body mass index; WC = Waist circumference; MBP = Mean blood pressure; TC = Total cholesterol; HDL-C = High density cholesterol; LDL-C = Low density cholesterol; TG = Triglycerides; CRF = Cardiorespiratory fitness; LVM = Left ventricular mass; * = $p < 0.005$.

The space visualisation of the network is presented in Figure 1. The blue color expresses positive relationships, and negative relationships between the variables are expressed by the red color. The thickness and saturation of the colors indicate the weight of the relationship.

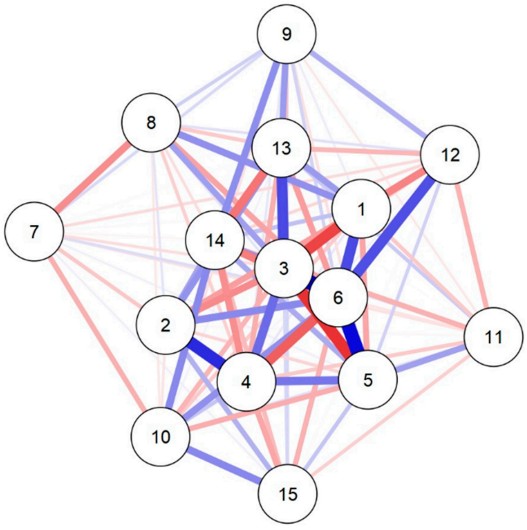

**Figure 1.** Network perspective of the relationship between moderate-to-vigorous physical activity, sedentary behaviour, and cardiovascular risk factors. Caption: 1 = Moderate-to-vigorous physical activity; 2 = Sedentary Behaviour; 3 = Age; 4 = Sex; 5 = Body mass index; 6 = Waist circumference; 7 = Mean blood pressure; 8 = High density cholesterol; 9 = Low density cholesterol; 10 = Triglycerides; 11 = Glucose; 12 = Fat; 13 = Cardiorespiratory fitness; 14 = Left ventricular mass; 15 = Sokolow–Lyon. Blue color lines = positive association; Red color lines = negative association.

Positive relationships were seen between MVPA and WC (0.499), HDL-C (0.307) and CRF (0.276). Conversely, MVPA was negatively associated with BMI (−0.251) and Fat (−0.341). For SB, positive associations were seen with WC (0.326), CRF (0.296), LVM (0.250) and Sokolow-Lyon (0.215) (Table 2).

**Table 2.** Weight matrix of variables MVPA, SB, age, sex and cardiovascular risk factors.

| Variables | MVPA | SB | Age | Sex | BMI | WC | MBP | HDL | LDL | TGL | Glucose | Fat | CRF | LVM | Sokolow |
|---|---|---|---|---|---|---|---|---|---|---|---|---|---|---|---|
| MVPA | 0.000 | | | | | | | | | | | | | | |
| SB | −0.267 | 0.000 | | | | | | | | | | | | | |
| Age | −0.513 | −0.319 | 0.000 | | | | | | | | | | | | |
| Sex | 0.062 | 0.581 | 0.408 | 0.000 | | | | | | | | | | | |
| BMI | −0.251 | −0.116 | −0.561 | 0.378 | 0.000 | | | | | | | | | | |
| WC | 0.499 | 0.326 | 0.716 | −0.444 | 0.664 | 0.000 | | | | | | | | | |
| MBP | 0.016 | −0.097 | −0.094 | −0.151 | 0.037 | 0.092 | 0.000 | | | | | | | | |
| HDL-C | 0.307 | 0.069 | 0.263 | −0.127 | 0.072 | −0.253 | −0.315 | 0.000 | | | | | | | |
| LDL-C | 0.047 | 0.008 | −0.174 | 0.065 | −0.086 | 0.122 | 0.105 | 0.129 | 0.000 | | | | | | |
| TG | −0.144 | −0.073 | −0.204 | 0.156 | −0.228 | 0.305 | −0.219 | −0.062 | −0.083 | 0.000 | | | | | |
| Glucose | −0.148 | −0.020 | 0.084 | −0.150 | 0.265 | −0.072 | −0.080 | 0.004 | −0.060 | 0.079 | 0.000 | | | | |
| Fat | −0.341 | −0.092 | −0.115 | 0.035 | 0.123 | 0.485 | −0.099 | 0.084 | 0.224 | 0.021 | −0.212 | 0.000 | | | |
| CRF | 0.276 | 0.296 | 0.503 | −0.223 | 0.164 | −0.271 | 0.011 | −0.146 | 0.277 | 0.031 | 0.144 | −0.199 | 0.000 | | |
| LVM | 0.171 | 0.250 | 0.318 | −0.311 | 0.237 | −0.355 | −0.027 | −0.089 | 0.314 | 0.329 | −0.217 | −0.108 | −0.386 | 0.000 | |
| Sokolow-Lyon | 0.203 | 0.215 | 0.110 | −0.248 | 0.158 | −0.216 | 0.038 | −0.093 | 0.112 | 0.331 | −0.128 | −0.012 | −0.075 | −0.187 | 0.000 |

MVPA = Moderate-to-vigorous physical activity; SB = Sedentary Behaviour; BMI = Body mass index; WC = Waist circumference; MBP = Mean blood pressure; HDL = High density cholesterol; LDL = Low density cholesterol; TG = Triglycerides; CRF = Cardiorespiratory fitness; LVM = Left ventricular mass.

The centrality indicators highlighted WC as the most important variable in the network, according to the Betweenness, Closeness, and Strength indices (Table 3).

**Table 3.** Centrality measures of the network analysis MVPA, SB, age, sex and cardiovascular risk factors.

| Variables | Network | | |
|---|---|---|---|
| | Betweenness | Closeness | Strength |
| MVPA | −0.502 | 0.585 | 0.480 |
| SB | −0.662 | −0.029 | −0.044 |
| Age | 0.459 | 1.506 | 1.628 |
| Sex | −0.182 | 0.596 | 0.576 |
| BMI | 0.139 | 0.504 | 0.574 |
| WC | 3.182 | 1.809 | 2.069 |
| Fat | −0.662 | 0.074 | −0.627 |
| MBP | −0.662 | −1.602 | −1.403 |
| HDL-C | 0.299 | −0.831 | −0.764 |
| LDL-C | −0.662 | −1.062 | −0.976 |
| TG | −0.182 | −0.211 | −0.510 |
| Glucose | −0.662 | −1.307 | −1.120 |
| CRF | −0.021 | 0.159 | 0.233 |
| LVM | 0.779 | 0.717 | 0.536 |
| Sokolow-Lyon | −0.662 | −0.908 | −0.651 |

MVPA = Moderate-to-vigorous physical activity; SB = Sedentary Behaviour; BMI = Body mass index; WC = Waist circumference; SBP = Systolic blood pressure; DBP = Dyastolic blood pressure; TC = Total cholesterol; HDL = High density cholesterol; LDL = Low density cholesterol; TG = Triglycerides; CRF = Cardiorespiratory fitness; LVM = Left ventricular mass.

## 4. Discussion

This study analyzed the association between different factors involved in obesity etiology, considering the synergic relationships between them as a network, in a sample of overweight low-income schoolchildren. Our main results showed a positive association between MVPA and HDL, CRF and WC.

It is well known that in general, overweight/obese children tend to spend less time on MVPA than their normal-weight peers [33]. Although the assessed children do not comply with the recommended 60 min/day of MVPA, our results are, at least partially, in line with

previous studies, as MVPA was positively associated with several cardiovascular outcomes. CRF is an important cardiovascular health indicator [34], negatively related to body fat [35], which mediates the association between CRF and clustered cardiovascular risk factors [36]. Cardiovascular benefits are seen at higher PA intensities [37], as higher intensities may provide sympathetic activity reduction, increased vasodilation, and consequently, increased vascular benefits when compared to low PA intensities [38]. Therefore, in children, CRF develops as a consequence of MVPA [39]. The observed positive association between MVPA and CRF is somehow expectable, as even in obese children this association may be seen [40].

Our results also showed a negative association between MVPA and BMI and body fat. Such results are already demonstrated in the literature, and MVPA interventions in obese children promote a decrease in body fat and also in BMI [41,42]. The explanation most mentioned in the literature is related to changes in energy balance, with an increase in energy expenditure and a decrease in caloric intake [43].

The results showed a positive association between SB and LVM and the Sokolow-Lyon index. The indicators of hypertrophy of the left ventricle represent a worrying variable, since obese children may be more prone to this condition. However, no study analysed the relationship between SB and LVM in this population, though the increase in the thickness of the left ventricular wall and the sedentary lifestyle have been identified as the main characteristics among children and young adult victims of sudden death [44]. It is important to consider that the high duration and frequency of SB are associated with metabolic markers with a higher cluster cardiovascular risk score [10].

Moreover, this study highlighted a positive association between MVPA and WC, SB and CRF. Regarding WC, some factors may partially explain this adverse association: firstly, PA may promote a compensatory effect, increasing caloric intake [45]; additionally, studies suggest that MVPA is positively associated with time spent in SB in children [46]. Ekelund et al. showed that positive changes in WC may be seen only for those children who spend longer MVPA periods [47].

Regarding the positive association between SB and CRF, although its association is most often verified with MVPA it has been observed in association with time in sedentary activities [48] and when the time in SB is not prolonged [49]. Again, this situation can also be the result of the compensatory effect, since the CRF is improved with more intense BP.

The cardiovascular risk factors and obesity, especially in childhood, present confusing relationships, since several factors can influence these associations, such as the period of intrauterine life [50], genetic aspects [51] and even eating behaviour [43].

Furthermore, it is important to consider that the analysis used in the present study allows us to verify the associations that occur synergistically. Thus, as previously described in the literature, MVPA showed an inverse association with SB, which must be considered when analysing the other associations. Previous studies have shown that combining these behaviours is one possibility to understand children's cardiovascular risk factors better [47,52,53]. A longitudinal study published by Hjort et al. (2014) observed that children with high MVPA/low SB had a better cardio—metabolic profile than children with low MVPA/high SB [52]. However, studies that consider these behaviours in combination with cardiovascular risk factors in children are still few and of low quality [53].

The network system also showed WC as the most important variable in the network, reinforcing its importance as a useful and easy-assessing measure. A priori studies have shown that WC can be used to stratify risk groups in children [54,55]. In the study by Sonmez et al. [56], it was found that WC was associated with greater carotid artery thickness in obese children, while BMI was not. This situation demonstrates the importance of this variable in the detection of subclinical cardiovascular diseases, including left ventricular hypertrophy [57]. This result is in line with the International Diabetes Federation when suggesting that due to the impossibility of detecting metabolic syndrome in children under 10 years old, WC should be considered as the main predictor of cardiovascular risk

factor [58], considering its strong association with insulin resistance, hyperlipidemia and blood pressure [59].

There are some limitations that the authors are aware of. The results of this study should be interpreted with caution, considering its limited sample, although the statistical analysis used is robust enough to support small sample sizes without compromising the results, and the Bootstrap resampling process was used, which minimises possible bias. Moreover, the assessment of several cardiovascular risk factors in a low-income overweight/obese child´s sample, and the use of objectively measures of MVPA, SB and LVM are strengths that should be highlighted.

The present study demonstrated that cardiovascular risk factors are interrelated, and showed that even in overweight low-income children, more time spent in MVPA and less time in SB are related to cardiovascular health. Interventions targeting cardiovascular health in this population should focus on reducing abdominal fat.

**Author Contributions:** Conceptualisation, T.B., A.S.F. and F.C.; methodology, T.B., A.S.F. and F.C.; formal analysis, N.Q., L.C. and P.B.; data curation, T.B., A.S.F., N.Q., L.C. and P.B.; writing—original draft preparation, T.B., A.S.F., N.Q., L.C., P.B. and C.R.; writing—review and editing, C.R., C.M. and F.C.; visualisation, C.M. and F.C.; supervision, F.C.; project administration, F.C. All authors have read and agreed to the published version of the manuscript.

**Funding:** No financial or material support of any kind was received for the work described in this article.

**Institutional Review Board Statement:** All the Helsinki Declarations' ethical aspects were followed (Association, 2013). The evaluation methods and procedures were approved by the Research Ethics Committee of Health Science Center (protocol n. 1.311.598) and by the Education Board of the city.

**Informed Consent Statement:** All children participating in the study had their parent/guardian's authorisation registered in the Consent Form. By signing the Consent Form, all parents/guardians of the children participating in the study authorised the publication of the data, as long as the identification and integrity of the children remained safe.

**Data Availability Statement:** Not applicable.

**Conflicts of Interest:** The authors have no conflict of interest relevant to this article.

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
