# Peer review of "Physical Activity, Sedentary Behaviour and Cardiovascular Risk Factors in Overweight Low-Income Schoolchildren: A Complex System Perspective"

_2673-4168, doi:10.3390/obesities3010008_

Round 1

Reviewer 1 Report

·      Your article seems to be fine and original.

·      Page 2, Lines 88: Please describe how was the sample size calculated?

·      Page 3, Lines 116-118: Please clarify the point you are attempting to make. "Three measurements 116 were performed, with a five-minute interval".

·      Page 4, Lines 157- 159: Provide a reference for this statement.

·      Page 9, Lines 253-254: Please clarify the point you are attempting to make. "The most pointed explanation in the literature is related to changes in the energy balance".

·      Page 9, Lines 267: Provide a reference for this statement. )Euklend et al.)

·      The idea behind this work is good but discussion needs precision and clarity. Could you elaborate more on this by specifically talking about analyze the relationship between (MVPA), (SB), and cardiovascular risk factors between boys and girls.

Author Response

Response to Reviewer 1 Comments

Point 1: Page 2, Lines 88: Please describe how was the sample size calculated?

Response 1: Dear reviewer, thank you for your comments. We added the description of the sampling calculation in lines 88 - 98.

Point 2: Page 3, Lines 116-118: Please clarify the point you are attempting to make. "Three measurements 116 were performed, with a five-minute interval".

Response 2: To meet such a request, we have detailed the points of this measure. Please see lines 130-132

Point 3: Page 4, Lines 157- 159: Provide a reference for this statement

Response 3: Done. See line 174, reference 26.

Point 4: Page 9, Lines 253-254: Please clarify the point you are attempting to make. "The most pointed explanation in the literature is related to changes in the energy balance".

Response 4: We add the explanation as requested on lines 268-270.

Point 5: Page 9, Lines 267: Provide a reference for this statement. (Euklend et al.)

Response 5: Done. Reference 47.

Point 6: The idea behind this work is good but discussion needs precision and clarity. Could you elaborate more on this by specifically talking about analyze the relationship between (MVPA), (SB), and cardiovascular risk factors between boys and girls.

Response 6: Again, we appreciate the suggestions. To discuss such behaviors in combination, we have added a paragraph to the discussion for this purpose. See lines 292-300.

Reviewer 2 Report

1. The study presents the results of original research.

2. Results reported have not been published elsewhere.

3. Experiments, statistics, and other analyses are performed to a high technical standard and are described in sufficient detail.

4. Conclusions are presented in an appropriate fashion and are supported by the data.

5. The article is presented in an intelligible fashion and is written in standard English.

6. The research meets all applicable standards for the ethics of experimentation and research integrity.

7. The article adheres to appropriate reporting guidelines and community standards for data availability.

Author Response

Point 1: The study presents the results of original research.

Point 2: Results reported have not been published elsewhere.

Point 3: Experiments, statistics, and other analyses are performed to a high technical standard and are described in sufficient detail.

Point 4: Conclusions are presented in an appropriate fashion and are supported by the data.

Point 5: The article is presented in an intelligible fashion and is written in standard English.

Point 6: The research meets all applicable standards for the ethics of experimentation and research integrity.

Point 7: The article adheres to appropriate reporting guidelines and community standards for data availability.

Thank you.
